# Some Mechanical Properties of Composite Materials with Chopped Wheat Straw Reinforcer and Hybrid Matrix

**DOI:** 10.3390/polym14153175

**Published:** 2022-08-03

**Authors:** Dumitru Bolcu, Marius Marinel Stănescu, Cosmin Mihai Miriţoiu

**Affiliations:** Department of Mechanics, University of Craiova, 107 Calea Bucureşti, 200512 Craiova, Romania; dbolcu@yahoo.com (D.B.); miritoiucosmin@yahoo.com (C.M.M.)

**Keywords:** composite materials, chopped wheat straw, hybrid resin, mechanical properties

## Abstract

Modern agriculture produces a very large amount of agricultural waste that remains unused. The use as a reinforcer of these renewable resources for the realization of composite materials, and the finding of useful industrial applications, constitutes or provokes the groups of researchers in this field. The study conducted in this article falls in this direction. Composites were fabricated with the chopped wheat straw reinforcement and epoxy resin matrix or hybrid resins with 50% and 70% Dammar volume proportions. Some mechanical properties of this type of composite materials were studied based on tensile strength, SEM analysis, water absorption/loss, vibration behavior and compression strength. The strength–strain and strain–strain diagrams, the modulus of elasticity, the breaking strength and the elongation at break were obtained. Compared to the epoxy resin composition, those with 50 and 70% Dammar, respectively, have a 47 and 55% lower breaking strength and a 30 and 84% higher damping factor, respectively. Because the values of these mechanical properties were limited, and in practice superior properties are needed, sandwich composites were manufactured, with the core of previously studied compositions, to which the outer faces of linen fabric were applied. These composites were applied to the bend (in three points), obtaining the force–deformation diagrams. The obtained properties show that they can be used in construction (paneling, shells, etc.), or in the furniture industry.

## 1. Introduction

Reinforced polymer composite materials are widely used in the industrial field for their excellent properties that allow for superior fatigue behavior over the lifetime [1]. Existing research has focused on the long-term evolution of single-type fiber-reinforced polymer composites, such as the glass fiber-reinforced polymer, GFRP [2], carbon fiber-reinforced polymer, CFRP [3], and hybrid fiber-reinforced polymer, HFRP [4].

In recent years, considerable attention has been paid to the production of new composite materials whose reinforcement materials are made from the recycling of waste or by-products in the agricultural-food industry. This increases the economic value and environmental benefits for their use in industrial applications [5,6,7]. The use of straw biomass in composites is increasing due to cost efficiency and the fact that they are lightweight, have low density and have a lower impact on the environment during production [8]. To date, the most commonly used material for the manufacture of organic composites is wood [9], but wheat straw as a renewable material has the potential to successfully replace wood in various applications. In general, wheat straw has chemical components similar to those of wood; more precisely, wheat straw contains 33.4–41.93% cellulose, 25.2–30.2% hemicellulose and 14.55–18% lignin, respectively [10,11,12].

Wheat straw is a by-product of the dry wheat stalk that remains after the grain and husk have been removed. Such natural fibers are extremely cost effective, with high specific characteristics of low-density fibers, and unlike other reinforcing fibers, they are biodegradable and non-abrasive [13]. Wheat straw, abundantly available, is currently used in the production of paper, ethanol, compost, absorbent materials, roofing materials and animal feed [14] but also in the automotive industry [15,16].

The use of wheat straw for industrial purposes is also useful due to the fact that the yield of wheat straw is about 1.3–1.4 kg for each kg of grain produced [17]. Therefore, a lot of wheat straw is plowed, or mostly burned in open fields. The smoke emitted during the burning of crop residues, under the effect of air circulation, can travel significant distances, thus carrying with it various pollutants, causing severe fog in the local, regional and global atmosphere [18]. Moreover, although open burning is conducted mainly in rural areas, under the effect of air circulation, its effects can be felt in remote areas, as well as in neighboring provinces and countries [19]. The incineration of straw in the open air not only wastes valuable renewable natural resources but also causes environmental pollution and poses major threats to the traffic and safety of highways, railways and civil aviation flights and seriously harms human health and safety [20]. Therefore, a number of alternative methods for straw use have been developed: it can be used as cattle feed [21], for energy production [22] and for composting and incorporation back into the soil [23]. An alternative method of removing wheat straw by calcination in a controlled environment and using the resulting ash as a substitute for cement in cement-based composites is presented in [24]. Studies on the use of agricultural ash, such as wheat straw ash (WSA) in concrete mixtures, are presented in [25]. The use of 10% (WSA) was considered to be optimal in terms of mechanical performance and durability [26] and therefore the use of agricultural ash in concrete mixtures is limited. Other possibilities to create the thermal properties of composite materials by inserting cellulose crystals and carbon fillers are presented in [27,28,29].

For the production of composite materials, agricultural waste can be used in combination with plastics such as: polyethylene terephthalate (PETE), polystyrene (PS), polypropylene (PP), polyvinyl chloride (PVC), low-density polyethylene (LDPE) and high-density polyethylene (HDPE) [30]. The characteristics of the composites obtained depend on the polymers used and the nature of the reinforcing materials but also on the manufacturing processes (extrusion, injection moulding and compression moulding). One of the main impediments to obtaining injection-moulded wheat straw-reinforced composites is that relatively short fibers can be used [31]. A disadvantage of using wheat straw is the relatively weak surface properties of wheat straw which can reduce the quality of the interfacial bond with different polymeric binders. In addition, straw fibers contain foreign substances, which could inhibit the ability to bind to the polymer matrix. This has ultimately hampered the commercialization of wheat straw composites [32]. Numerous studies have been performed to improve the adhesion between the straw fibers and the matrix by modifying the fiber by physical, mechanical and chemical methods.

The mechanical behavior of composite materials is influenced by the mechanical properties of the matrix and the fibers, the volume ratio, the orientation and the load of the fibers but also the interfacial connection between the matrix and the fibers [33]. In order to produce high-performance bio-composites, several methods have been developed to improve the compatibility of natural reinforcement surfaces, by additive incorporation or by pre-treatment processes. Therefore, a surface treatment of wheat straw is necessary to reduce the hydrophilicity, which allows the formation of a strong bond between the resin and the straw, resulting in better mechanical properties and increased thermostability. Fiber surface treatment mainly includes acetylation, silane and mercerization [34], hydrothermal extraction, rays or plasma [35,36], enzymatic [37,38] and steam [39]. Alkaline pre-treatments [40,41] have been shown to be beneficial in changing the surface of the straw with the partial removal of extracts and waxes, which has made them more hydrophilic and more compatible with water-based resins. The microcellular structure of the straw has changed and, as a result, the penetration of the matrix into the cellular lumen has been improved, leading to an intimate connection between the substrate and the matrix [42,43]. Wheat straw treated with 2% sodium hydroxide has been found to have better physicochemical and mechanical properties but increasing the alkaline concentration can damage the fiber surface, leading to a decrease in mechanical performance [44].

The mechanical properties of wheat fiber bundles are inferior to the properties of traditional textile fibers. Due to the very low bending capacity, the use of wheat straw for some technical applications is reasonable only in the form of a whole stalk. The presence of nodes can alter the mechanical properties of wheat straw-reinforced composites. The node has a very different microstructure from that of the internode portion [45]. It is structurally inferior, with low mechanical properties, and the node surface chemical inhibits the good interfacial bond between the substrate and matrix in bio-composites [32,45]. In order to improve its properties, in [46], the wheat straw was optimized by separating the nodes, and the internode portions were subjected to hybrid pre-treatments with hot water, followed by steam, for two different lengths of time and drying in a microwave oven.

Straw fibers have been used to make bio-composite boards (BCBs) that have great potential for replacing synthetic composites, which could lead to cheaper, more durable and environmentally friendly materials [47]. Ways to improve mechanical properties for a BCB, using different types of resins, are presented in [48]. The use of natural resins in combination with wheat straw creates new prospects for obtaining totally renewable materials. Dammar is one of the natural resins that formed the basis of hybrid resins. The chemical and mechanical properties of such a resin have been studied in [49]. This type of hybrid resin has been used to make composite materials with various types of natural reinforcers [50,51]. These vibration behaviors for composite materials are also studied in these works. The study of water absorption for hybrid resin composition based on Dammar reinforced with chopped seed husks was studied in [52].

In this paper, the mechanical behavior of composite materials with a hybrid resin matrix based on Dammar and chopped wheat straw reinforcer are studied. Using tensile and compression tests, some mechanical properties such as modulus of elasticity, tensile strength and elongation at break were determined. The damping coefficient and the own frequency for the specimens from the realized sets were determined experimentally and the loss factor for the obtained materials was established. Water absorption/loss has also been studied. If these composite materials underwent a flexural strength, they would have limited mechanical properties. To remove this disadvantage, they were used as the core of sandwich composites, with outer faces made of natural fiber fabrics impregnated with the same type of resin as the core.

## 2. Materials and Equipment Used

### 2.1. Resins Used

Three types of resin were used to make the composite materials:-Resoltech 1050 epoxy resin and the corresponding Resoltech 1055 hardener, the technical data of which can be found on the manufacturer’s website (see [53]); this resin will be abbreviated E;-Hybrid resin with a volume ratio of 50% natural Dammar and 50% Resoltech 1050 epoxy resin with the corresponding hardener; this resin will be abbreviated D5;-Hybrid resin with a volume ratio of 70% natural Dammar and 30% epoxy resin of the Resoltech 1050 type with the corresponding hardener; this resin will be abbreviated D7.

The chemical composition of D5 and D7 resins was determined by EDS analysis. This analysis was performed with the QUANTA INSPECT F50 scanning electron microscope (see [54]) provided with:-Field emission gun (FEG), with a resolution of 1.2 mm and energy dispersive X-ray spectrometer (EDS), with a MnK resolution of 133 eV;-EDAX chemical micro-composition analyzer and its related software for performing local micro-composition analyses.

The chemical and mechanical properties of these hybrid resins were investigated in detail in [49], respectively, [51]. The use of hybrid resins is necessary because natural resins diluted with solvents form varnishes that require hardening to combine with synthetic resins [55]. The volume ratio of Dammar is limited between 50 and 70% because it has been found that, for volume proportions higher than or equal to 80% natural Dammar resin, the curing time of the hybrid resin increases greatly and therefore the hybrid resin is no longer “interesting” in terms of applications in the field of composite materials. For these reasons, the limits of 50 and 70% Dammar in the hybrid resin were used in this study.

### 2.2. Tensile Test

Using the matrix of the three types of resin specified above, three plates of composite materials with chopped wheat straw reinforcement were cast. From each manufactured plate, using a water-jet-cutting machine, a number of 15 samples were cut to be used for the tensile test. The test pieces were 250 mm long and 25 mm wide in accordance with ASTM D3039 [56]. The required tensile specimens were abbreviated as follows: PTE.1-15 composite specimens with epoxy resin matrix, PTD5.1-15 composite specimens with hybrid resin matrix D5 and PTD7.1-15 composite specimens with hybrid resin matrix D7 (see Figure 1).

The main characteristics of these specimens are presented in Table 1.

The casting and polymerization process were performed at a controlled ambient temperature of 21–23 ${}^{{}^{\circ}}$C. In order to ensure the hardening of the plates, a complete polymerization, the specimens with the epoxy resin matrix were cut 5 days after casting, and the specimens with the hybrid resin matrix were cut 10 days after casting.

The tensile strength of the PTE, PTD5 and PTD7 specimens was achieved by means of the LLOYD Instruments Lrx PLU mechanical test machine, equipped with bending bodies for 3-point testing, with a maximum force of 2.5 kN, a maximum cross member stroke of 1400 mm and a 50 mm extensometer (see [57]). A tensile strength rate of 5 mm/min was used for the tensile test.

The study of the fracture surfaces was carried out, after the tensile test, in the cross section of the PTE, PTD5 and PTD7 specimens, using an Olympus SZX7 Stereo Microscope, SZ2-ET with Galilean optical system, with a resolution of up to 600 lines per millimeter and a 7:1 zoom ratio (see [58]). The analysis was performed in accordance with ASTM STP 1203 [59].

### 2.3. Water Absorption

The water absorption of one specimen in each PTE, PTD5 and PTD7 set was studied using a Kern ABJ 220-4NM analytical balance with single-cell technology and a weighing accuracy of 0.0001 g (see [60]); a polypropylene laboratory tray measuring 375 × 300 × 75 mm; 2000 mL drinking water and aluminium foil. The three specimens used for this study had the same dimensions as those used to request traction.

### 2.4. Vibration Behavior

The study of material damping is important because it significantly influences the behavior of vibrating systems, especially when operating close to resonant frequencies. The experimental study is necessary because the damping properties of many new materials are not yet properly known. The energy dissipation mechanisms in fiber-reinforced composites mainly include the viscoelastic characteristics of the matrix and fibers, the damping due to the interface region and the cracks of the matrix and the viscoplastic damping [61]. In general, three energy dissipation mechanisms are considered that consider that the damping force is proportional to the velocity, the vibration frequency or the square of the vibration frequency.

The damping properties were studied for composites reinforced with short coir [62], bananas [63], flax and hemp [64], hybrid fiber ramie/glass [65], Luffa cylindrica [66] or Sansevieria cylindrica [67]. The damping properties of Dammar-based hybrid resin composite materials reinforced with linen, cotton or hemp fabrics are studied in [50], and those reinforced with waste paper are studied in [51]. The results of these analyses indicated that the loss factor is determined by the shape of the reinforcing fibers and the orientation of their fibers significantly affects the damping behavior. It has been observed that damping can be improved with a slight decrease in the rigidity of the composite by generating a larger number of interfaces.

A SPIDER 8 data acquisition system, connected to the NEXUS 2692-A-0I4 signal conditioner and a 0.04 pC/ms${}^{-2}$ sensitivity accelerometer, was used to study the vibration behavior. CATMAN EASY software for data acquisition and processing was used for data acquisition and processing. The three specimens used to study the vibration behavior had the same dimensions as those used for tensile strength.

### 2.5. Compressive Test

For the compressive test, using the same components, three bars with a square section were cast, with the side of the section measuring 21 mm. The mass proportion of chopped wheat straw was: 42% for the epoxy resin bar, 42% for the D5 hybrid resin bar and 41% for the D7 hybrid resin bar. The densities of the materials obtained were: 0.97 g/cm${}^{3}$ for the epoxy resin bar, 0.93 g/cm${}^{3}$ for the D5 hybrid resin bar and 0.92 g/cm${}^{3}$ for the D7 hybrid resin bar. A total of 15 cubes were cut from each bar (using a disk flow machine) in accordance with ASTM D3410/D3410 M ([68]). These were abbreviated as follows: PE.1-15 composite specimens with epoxy resin matrix, PD5.1-15 composite specimens with hybrid resin matrix D5 and PD7.1-15 those made of resin matrix composite hybrid D7 (see Figure 2).

The compressive test of PE, PD5 and PD7 specimens was made with the Walter-Bai LF300 universal static and dynamic test machine, with the capacity of 300kN and having the following technical characteristics: ±300 kN static force, ±250 kN dynamic force, 250 N/s speed load and the frequency of load cycles up to and including 20 Hz.

### 2.6. Flexural Test

In the second stage, three plates of the same composite materials were cast, as in the case of the tensile strength. The mass proportion of chopped wheat straw was: 38% for epoxy resin plate, 38% for D5 hybrid resin plate and 39% for D7 hybrid resin plate. The plates had the following thicknesses: 12.9 mm epoxy resin plate, 12.9 mm D5 hybrid resin plate and 12.8 mm D7 hybrid resin plate. On both sides of these plates were applied 4 layers of linen fabric impregnated with the same type of resin as the plate. After applying the outer layers, the thickness of the sandwich plates was: 14.9 mm for the epoxy resin plate, 14.9 mm for the D5 hybrid resin plate and 14.8 mm for the D7 hybrid resin plate. The densities of the sandwich plates obtained were: 1.02 g/cm${}^{3}$ for the epoxy resin core plate, 0.99 g/cm${}^{3}$ for the D5 hybrid resin core plate and 0.98 g/cm${}^{3}$ for the D7 hybrid resin core plate.

Ten test specimens were cut from each plate. The dimensions of these test specimens were 150 mm long, 20 mm wide. These specimens were used for the flexural test and are abbreviated as follows: SPE.1-10 for composite specimens with epoxy resin matrix; SPD5.1-10 for composite specimens with D5 hybrid resin matrix; SPD7.1-10 for D7 hybrid resin matrix composite specimens (see, for example, the Figure 3). The (3-point) flexural test was performed according to ASTM C393-C393M-06 [69].

The request was made with the help of LLOYD Instruments Lrx PLU mechanical testing machines, as in the case of tensile test. The force–strain diagram was obtained from the maximum force $F_{max}$(N) and the corresponding maximum extension *e* (mm).

## 3. Results and Discussions

### 3.1. The Chemical Composition of the Resins Used

Two specimens were taken from the two hybrid resins. As the resin samples were non-conductive and the scanning electron microscope could not record micrographs on them, they were prepared before being subjected to an EDS analysis. More precisely, they were placed for 60 s in a metal provided with a gold target in which a vacuum was created, and a very small amount of gold was deposited to show conductivity. This process does not influence the final result of the chemical composition, it is only a gold coating to be able to perform the analysis (at most, gold is also identified as an element in the EDS analysis). The specimens were analyzed with the EDAX detector which gave the graphical representation of the atomic spectrum distribution of the identified elements and the numerical values of the chemical composition of the D5 and D7 resins.

The investigation of the morphological characteristics related to each specimen was performed both on the surface and in sections.

Figure 4 shows graphically the distribution of the atomic spectra of the elements identified in the D7 hybrid resin, obtained at a 2.110 keV intensity. The general spectrum obtained highlights the presence of the elements carbon, nitrogen, oxygen, chlorine, potassium and calcium in proportions that are found in Table 2.

### 3.2. Traction Load

The characteristic curve was obtained by determining the modulus of the elasticity, tensile strength and elongation percentage after breaking. Figure 5 shows the characteristic curves (strength–strain diagrams) for representative specimens from the PTE, PTD5 and PTD7 sets.

Table 3 shows the mean value and the mean square deviation for the determined mechanical properties. For the calculation of these values, the formula from [52] was used.

The tensile behavior of the studied composites depends on the proportion of the natural Dammar resin in the hybrid resin used as a matrix. The highest tensile strength was obtained for the composite materials with an epoxy resin matrix. However, it is lower than the tensile strength of the epoxy resin used (14.58 MPa compared to 76.1 MPa for the Resoltech 1050 epoxy resin (see [53])). This decrease can be explained by the fact that the chopped straw has irregular shapes, with sharp corners, which are primers for the propagation of cracks and produce stress concentrations. For materials with a hybrid resin matrix, the average tensile strength of the PTD5 and PTD7 specimens is 53 and 45%, respectively, of the average tensile strength of the PTE specimens. Moreover, in the case of the hybrid resin composites, the tensile strength is lower than the tensile strength of the resins used [51]. On the other hand, for the modulus of elasticity of the studied composite materials, there is an increase compared to the modulus of elasticity of the resins used. For the PTE composites, the average modulus of elasticity is 4652 MPa compared to 3500 MPa for the Resoltech 1050 epoxy resin [53]; for the PTD5 and PTD7 composites, the average modulus of elasticity is 3140 and 1684 MPa, respectively, compared to 1720 and 1260 MPa, which are the moduli of elasticity of the hybrid resins used [51].

Based on the stereo microscopic analysis (SEM), in Figure 6, images with the breaking surface of some representative specimens from the three sets, PTE in Figure 6a, PTD5 in Figure 6b and PTD7 in Figure 6c, are presented.

The rupture of the specimens was sudden, without the occurrence of the flow phenomenon. This type of phenomenon is characteristic of fragile materials, where the rupture takes place at the interface between the matrix and the reinforcer. From the analysis of Figure 6a, it can be seen that the rupture zone has a smooth appearance. This shows that in the case of the PTE specimen (with the epoxy resin matrix), the rupture took place without tearing the wheat straw from the matrix. In the case of Figure 6b,c, which show the breaking area for the PTD5 and PTD7 (hybrid matrix) specimens, the breaking areas are rougher and show fragments of wheat straw torn from the matrix, meaning there is good adhesion of the reinforcing material to the matrix. This phenomenon occurs because the polymerization time of the hybrid resin is much longer than that of the epoxy resin, and during all this time, the wheat straw has time to be impregnated with the hybrid resin.

### 3.3. Water Absorption

The study of water absorption took place in a room with a controlled temperature of 22–23 ${}^{{}^{\circ}}$C, lasted 9 days and went like this: the specimens were placed in a tray, drinking water was poured over them and the tray was covered with aluminium foil. The weighing process took place daily at the same time. It is found that after 7 days, the water absorption was negligible (less than 0.05 g). Figure 7 shows the variation of the water absorption in percentage of the three studied specimens.

Regarding the total water absorption of the studied specimens, it can be seen that it was 8.04% for the PTE specimen, 10.69% for the PTD5 specimen and 13.60% for the PTD7 specimen. One can notice that with the increase in the proportion of the Dammar resin in the composite matrix, there is also an increase in the water absorption. Three specimens of the three types of resin used were also tested for the study of water absorption. It was found that their water absorption was negligible (less than 0.1% of the mass of the test piece). Therefore, in the case of the composite materials, the water absorption is due only to the reinforcer (chopped wheat straw).

### 3.4. Vibration Behavior

The modulus of elasticity and the loss factor are material characteristics, while the frequency and damping coefficient are influenced by the bar size, boundary conditions and material characteristics. For these reasons, the frequency and damping coefficient are determined experimentally. For one test piece in the PTE, PTD5 and PTD7 sets, embedded at one end, the vibration was recorded at the free end. Measurements were made for free lengths of 120, 140, 160, 180, 200 and 220 mm.

Figure 8 shows an experimental vibration recording and how to determine the damping factor and the vibration frequency for the PTD7 test piece and the free length of 120 mm. This recording was chosen because it is the one with the highest vibration damping factor. The determination of the damping factor per unit mass of the bar and of the vibration frequency was made with the methods presented in [50,70].

Figure 9 and Figure 10 show variations in vibration frequency and damping factor, depending on the length of the bar console, for the representative specimens of the three types of composite materials reinforced with chopped wheat straw.

One observes that the change of the damping factor and the vibration frequency have similar variations, depending on the length of the bar. It can be appreciated that, in the case of the studied composites, among the presented damping mechanisms, the predominant one is the one in which it is considered that the damping is proportional to the vibration frequency. The damping factor characterizes the vibration damping capacity for the studied bars and depends on the length of the vibrating bar. Therefore, the damping factor depends both on the damping properties of the constituent materials but also on the dimensions of the bars and is therefore an element that characterizes the overall vibration damping capacity for the test specimens under study. In order to assess the damping capacity of the materials from which the test pieces are made, the loss factor is determined, with the relation (see [50]):(1)$$\eta =\frac{\mu }{\pi \nu }.$$

The average value of the loss factor for the studied composite materials is:-η=0.0259 for composite PTE;-η=0.0338 for composite PTD5;-η=0.0476 for composite PTD7.

The values of the loss factors for the three studied materials show that the composite materials with a hybrid resin matrix have damping properties superior to those with an epoxy resin matrix. Thus, the loss factor for the D5 hybrid resin composite is 30% higher than for the epoxy resin composite. For the D7 hybrid resin composite, the loss factor is 84% higher than for the epoxy resin composite.

### 3.5. Compressive Strength

The PE, PD5 and PD7 test specimens were applied a compressive strength. Consequently, the force–strain diagram was obtained from which the maximum force $F_{max}$ (N), the maximum strain $L_{max}$ (mm) and the allowable compressive strength $\sigma_{ac}$ (MPa) were determined. Figure 11 shows the force–strain diagrams for the representative specimens from sets PE, PD5 and PD7.

The mean value and the mean square deviation for the maximum load force $F_{max}$ (N) and the maximum deformation $L_{max}$(mm) for the sets of the specimens of type PE, PD5 and PD7 are given in Table 4.

For this strength, too, the behavior of the studied composite materials is similar to that of the tensile strength. The highest values for the maximum load and maximum strength were obtained for the epoxy matrix composites. For the D5 and D7 matrix composites, the maximum compressive strength is 75 and 43% of the maximum strength on the epoxy matrix composites. If, in the case of the tensile test, the elongation at break has a slight increase in hybrid resin composites, in the case of the compressive strength of the deformations of the hybrid resin matrix composites, the deformations are much larger than in the epoxy resin matrix composites. Thus, for the composites PD5 and PD7, the deformation at the maximum force is 64 and 104%, respectively, higher than for the PE composites.

### 3.6. Flexural Strength

The Dammar resin is homogenized with the synthetic resin, and the increase in the volume proportion of the Dammar decreases the values of some of the mechanical properties of the hybrid resin and implicitly of the composite materials that use these resins as a matrix. For these properties, the studied materials were used as a core for sandwich boards with the outer faces made of natural fiber fabrics, impregnated with the same type of matrix.

Figure 12 shows the force–extension diagrams for the representative specimens from the SPE, SPD5 and SPD7 sets.

Table 5 shows the mean value and the mean square deviation for the maximum load force $F_{max}$ (N) and the corresponding maximum extension *e* (mm) for the SPE, SPD5 and SPD7 test specimen sets.

The diagrams in Figure 12 show that the studied sandwich composites have a mechanical behavior similar to that which was observed during the tensile and compressive tests. Specifically, if the maximum load is obtained on the SPE specimens, then on the SPD5 and SPD7 specimens, respectively, this load represents 60 and 47.5% of the maximum SPE load, respectively. In the case of the maximum extension, the lowest value is recorded for the SPE specimens, and for the SPD5 and SPD7 specimens, this extension is 15 and 38% higher, respectively, than for the SPE specimens.

In all the mechanical tests studied, in terms of the tensile strength, for the specimens made with D5 and D7 resins, respectively, lower values were obtained than those obtained for the specimens made with epoxy resin. Taking into account the results obtained for the composite materials made of epoxy resin, the composite materials made of hybrid resin with 70% Dammar had a behavior proportional to all three mechanical tests (tensile, compressive and flexural). In contrast, the composite materials made with 50% Dammar hybrid resin behaved better when submitted to compressive than when submitted to tensile testing. An explanation for this behavior is that, in its natural state, Dammar is friable, yielding very easily to stretching stresses. Consequently, increasing the volume proportion of Dammar above 50% in the hybrid resin makes it have a mechanical behavior similar to Dammar. In the case of the flexural test, it is found that the specimens performed better in this case than in the tensile test and worse than in the compressive test. This is because the lower part of the specimens is under the flexural strength and the upper part is under the compressive strength.

The use of Dammar-based hybrid resins as a matrix has some advantages over synthetic resins, for example:-A much lower cost price (the cost of Dammar is 4–5 times lower than that of epoxy resin);-Dammar is non-toxic and does not pose a danger to human health (it is used in the medicine industry for the treatment of stomach diseases);-Dammar is biodegradable.

There are some advantages in using chopped wheat straws as reinforcing material:-They are a by-product of agriculture, and therefore biodegradable;-They are produced regularly and are found in abundance in nature;-They are an alternative to the use of wood.

## 4. Conclusions

The increase in the proportion of the Dammar natural resin in the matrix of composite materials generates a significant change in their mechanical behavior:-In the case of tensile strength, the value of the modulus of elasticity and tensile strength decreases with the increasing volume of Dammar, and the elongation at break increases with the increasing volume of Dammar.-The analysis of the breaking sections and of the characteristic curves shows that a fragile rupture takes place, which is made suddenly, without the appearance of plastic deformations and without the existence of a flow zone. In the case of the composite materials with a hybrid matrix, a type of rupture occurred in which the matrix detached from the fibers (the fibers were torn from the resin), whereas in the case of the composite materials with an epoxy matrix, a type of rupture appeared in which the rupture breaks the fibers and the matrix, which maintains contact with the straw in the area where the rupture occurred (in a direction perpendicular to the direction of stress).-The modulus of elasticity for all the tested specimens has higher values than the modulus of elasticity of the matrix; this shows that regardless of the orientation of the reinforcer, there is an increase in the rigidity of the studied specimens.-The compressive strength shows that the maximum loading force decreases as the proportion of Dammar increases, while the maximum deformation increases as the proportion of Dammar increases.-Because all the particles in a section of the composite material vibrate equally regardless of the direction in which the straws in that section are oriented, the damping coefficient does not depend on the orientation of the fibers. Increasing the proportion of Dammar in composite materials reinforced with chopped wheat straw leads to a decrease in values for some mechanical properties, such as the tensile strength and modulus of elasticity, but allows for the production of industrial products with a high vibration damping capacity.

Sandwich composite materials with Dammar-based hybrid resin matrix, natural fiber fabric surfaces and shredded wheat straw core can be used in construction (for paneling, formwork, etc.) or in the furniture industry as an alternative to the use of chipboard. This combination leads to finished products with increased rigidity.

## Figures and Tables

**Figure 1 polymers-14-03175-f001:**
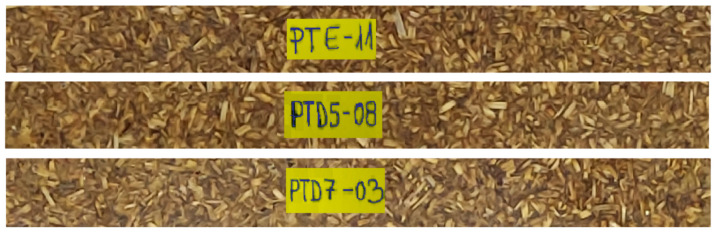
Test pieces submitted to tensile test.

**Figure 2 polymers-14-03175-f002:**
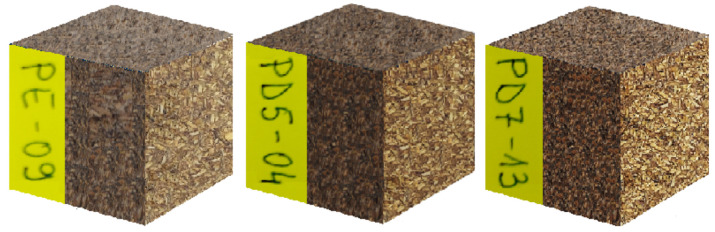
Cubes submitted to compressive test.

**Figure 3 polymers-14-03175-f003:**
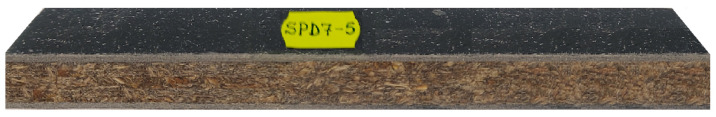
Sandwich composite test specimen with D7 hybrid resin matrix, which has a chopped wheat straw core and flax fabric faces, used for flexural test (in 3 points).

**Figure 4 polymers-14-03175-f004:**
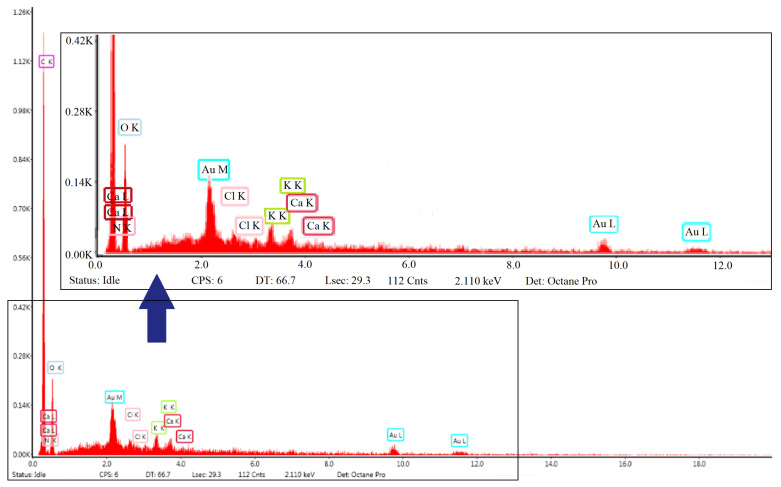
Distribution spectrum of the chemical elements in specimen D5 by EDS analysis, obtained at a 2.110 keV intensity.

**Figure 5 polymers-14-03175-f005:**
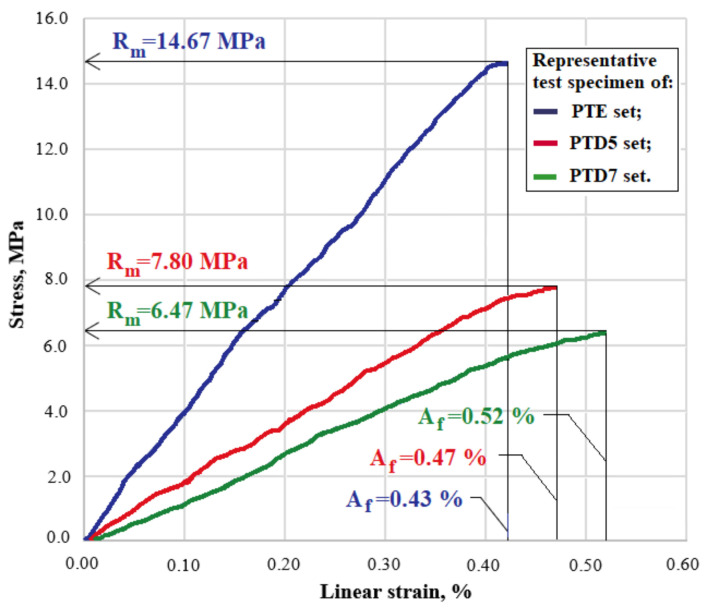
Strength–strain diagram for representative specimens from PTE, PTD5 and PTD7 sets.

**Figure 6 polymers-14-03175-f006:**
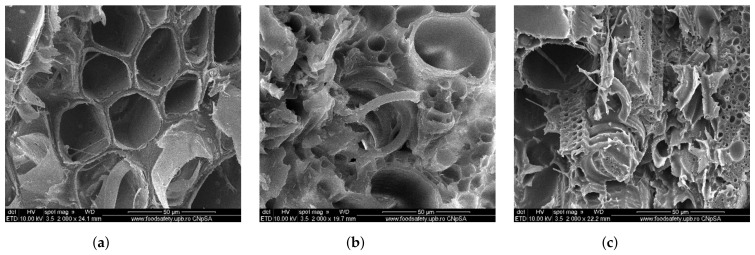
Breaking surface of some representative specimens from the sets: PTE (**a**); PTD5 (**b**); PTD7 (**c**).

**Figure 7 polymers-14-03175-f007:**
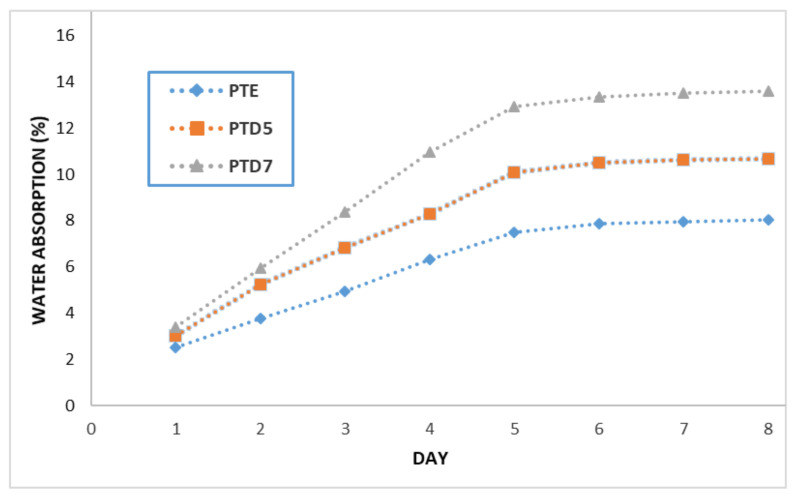
Variation of water absorption in percentage of PTE, PTD5 and PTD7 specimens.

**Figure 8 polymers-14-03175-f008:**
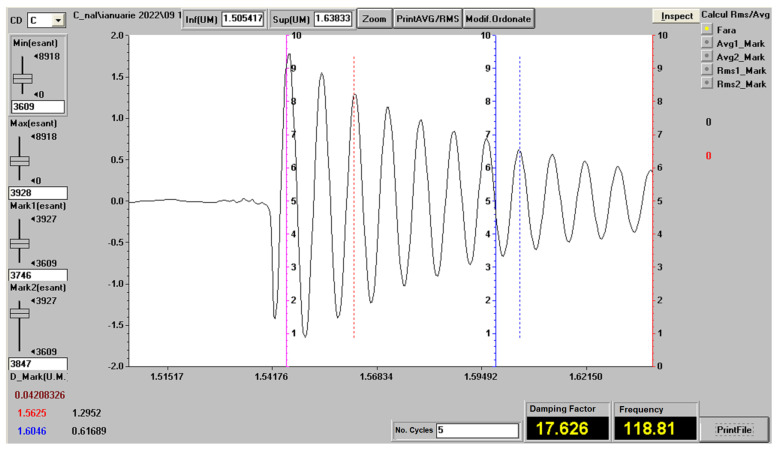
Determination of the frequency and damping factor for the test piece in the PTD7 set and the free length of 120 mm.

**Figure 9 polymers-14-03175-f009:**
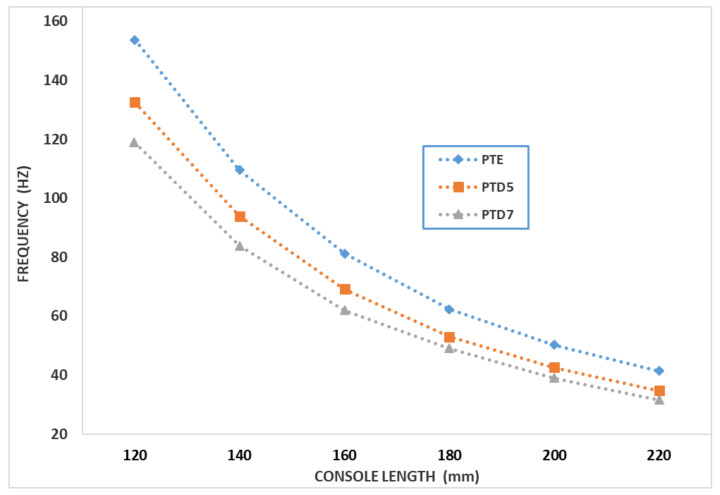
Variation of the vibration frequency as a function of the length of the bar console, for representative specimens from the PTE, PTD5 and PTD7 sets.

**Figure 10 polymers-14-03175-f010:**
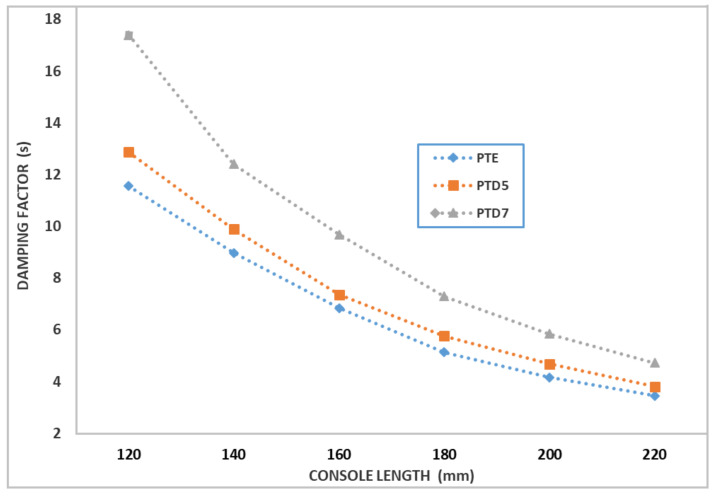
Variation of the damping factor according to the console length of the bar, for representative specimens from the PTE, PTD5 and PTD7 sets.

**Figure 11 polymers-14-03175-f011:**
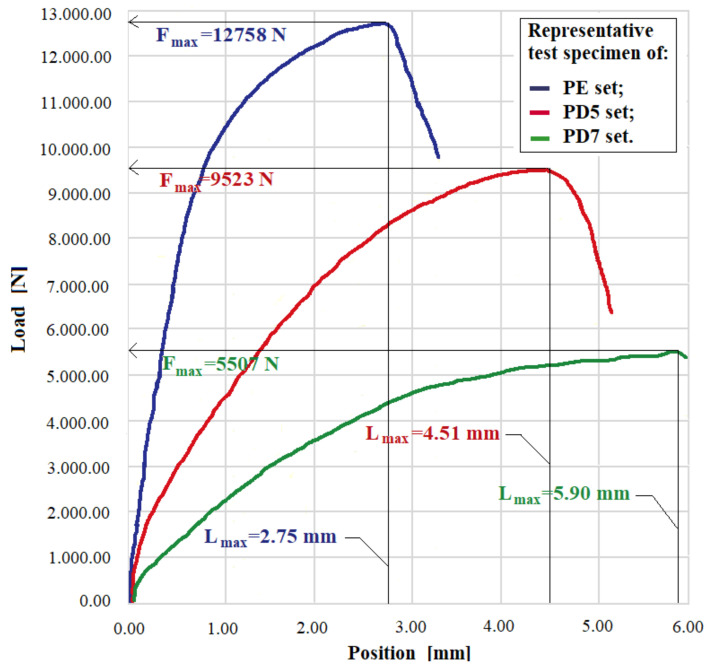
Deformation–force diagram for representative specimens from PE, PD5 and PD7 sets.

**Figure 12 polymers-14-03175-f012:**
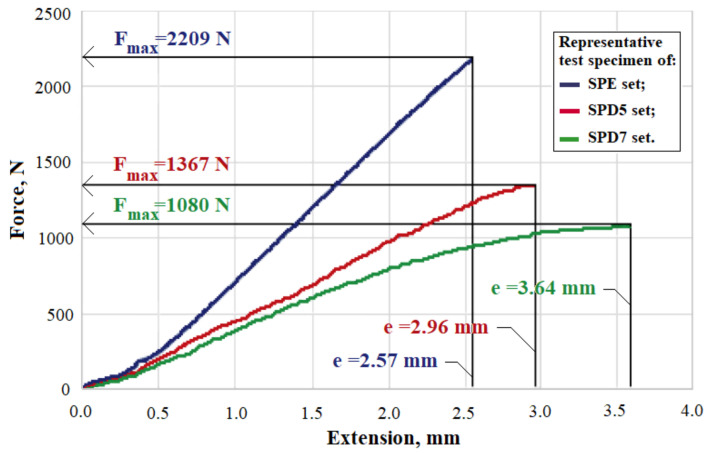
Strength–extension diagram for representative specimens from sets SPE, SPD5 and SPD7.

**Table 1 polymers-14-03175-t001:** Individual values of the chemical composition for the analyzed areas.

Resin	Mass Proportion	Thickness	Density
	of Wheat Straw (%)	Specimens (mm)	(g/cm${}^{3}$)
Epoxy	40	6.9	0.99
Hybrid D5	41	7.0	0.95
Hybrid D7	39	7.0	0.93

**Table 2 polymers-14-03175-t002:** Individual values of the chemical composition for the analyzed areas.

	Weight	Atomic	Error		K				
Element	(%)	(%)	(%)	Net Int.	Ratio	Z	R	A	F
C K	66.06	72.23	6.11	193.81	0.4357	1.0128	0.9924	0.6513	1
N K	4.69	4.4	99.99	1.87	0.0035	0.9924	1.0024	0.0751	1
O K	27.92	22.92	13.34	38.72	0.0286	0.9747	1.0115	0.1052	1
Cl K	0.3	0.11	26.63	8.85	0.0027	0.8383	1.071	1.0181	1.0282
K K	0.53	0.18	19.54	14.56	0.0049	0.8365	1.0809	1.0653	1.0476
Ca K	0.5	0.16	23.53	12.07	0.0048	0.8529	1.0854	1.0683	1.0528

**Table 3 polymers-14-03175-t003:** Mean value and mean square deviation for modulus of elasticity, tensile strength and elongation at break for PTE, PTD5 and PTD7 specimens.

Test Specimen	Modulus of Elasticity *E* (N/mm${}^{2}$)		Tensile Strength $R_{m}$ (MPa)		Elongation at Break *A* (%)	
Type	Average	Average	Average	Average	Average	Average
	Value	Square Deviation	Value	Square Deviation	Value	Square Deviation
PTE	4652	112	14.58	0.59	0.44	0.021
PTD5	3140	84	7.68	0.37	0.47	0.023
PTD7	1684	47	6.51	0.30	0.51	0.023

**Table 4 polymers-14-03175-t004:** Mean value and standard deviations for maximum load force $F_{max}$ (N) and maximum deformation $L_{max}$ (mm) for PE, PD5 and PD7 specimen sets.

Test Specimen	Maximum Load $F_{\max}$ (N)		Maximum Extension at $F_{\max}$ $L_{\max}$ (mm)		Compressive Strength $\sigma_{\mathrm{ac}}$ (MPa)	
Type	Average	Average	Average	Average	Average	Average
	Value	Square Deviation	Value	Square Deviation	Value	Square Deviation
PE	12,689	204	2.81	0.07	28.8	1.41
PD5	9490	163	4.62	0.11	21.5	1.02
PD7	5467	86	5.78	0.14	12.4	0.68

**Table 5 polymers-14-03175-t005:** Mean value and standard deviations for maximum load force and corresponding maximum extension for SPE, SPD5 and SPD7 test specimen sets.

Test Specimen	Maximum Load $F_{\max}$ (N)		Maximum Extension at $F_{\max}$ *e* (mm)		Flexural Strength $\sigma_{b}$ (MPa)	
Type	Average	Average	Average	Average	Average	Average
	Value	Square Deviation	Value	Square Deviation	Value	Square Deviation
SPE	2240	51	2.65	0.07	35.6	1.42
SPD5	1352	38	2.93	0.08	22.3	1.01
SPD7	1065	36	3.55	0.11	17.6	0.80

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
