# Peer review of "Some Mechanical Properties of Composite Materials with Chopped Wheat Straw Reinforcer and Hybrid Matrix"

_polymers, 2022, doi:10.3390/polym14153175_

Round 1

Reviewer 1 Report

The authors studied the mechanical properties, absorption properties and vibration behavior of three new green composites. Some results are interesting. However, there are logical relationship confusion in current writing. To further improve the quality of the paper and meet the needs of publishing, it is suggested to consider the following comments. 

1# The current abstract does not convey some key information. Some qualitative and quantitative results on the mechanical properties, vibration properties, water absorption properties of composites are suggested to be presented in the current content. According to the findings of this research work, it can further promote the engineering applications of green composite.

2# Introduction: this work aims to promote the engineering application of new green composite from agricultural waste in the field of industry as building materials. This idea is innovative, it can realize some advantages, such as waste utilization, ecological and environment protection, energy conservation and so on. Meanwhile, the above application of new green composite is also an attempt and challenge for synthetic fiber reinforced polymer composites, such as CFRP, GFRP, BFRP etc. As known, synthetic fiber composites have been widely used in the industrial field because of their excellent properties, mature preparation and application technologies. Therefore, to promote the new applications of green composites in the industrial field, this work should summarize some properties, advantages and application of synthetic fiber composites in the first part of introduction, and further put forward the main problems in terms of properties, cost, ecological environment to verify the future development potential of green composites in engineering applications. The authors can refer to the following latest work on performance, advantages and applications of synthetic fiber composites, such as Composite Structures, 2020; 246: 112418.  Construction and Building Materials, 2018, 161: 634-648.  Materials and Structures, 2020, 53: 73.

3# This paper studies the mechanical properties, microscopic properties, vibration behavior and water absorption properties of green composites. To be closely related to the current research content, the reviewer suggested that the authors should add the research on the vibration behavior and water absorption of green composites.

4# There is a serious logical confusion in current writing in part 2. The experimental test should be separated from the results and discussion to form two independent parts. Part 2 should focus on the raw materials, sample preparation and mechanical property tests. Part 3 should be the results and discussion.

5# Please explain why two hybrid resins systems are used in the experiment, and what advantages are there through the two hybrid resins? Will the two resin systems blend well? If it cannot be well blended, it is likely to form some initial defects owing to the poor interface.

6# Please provide some basic information about raw materials and sample preparation.

7# Figure 1 is unclear, it is suggested to enlarge the key spectrum.

8# It can be found in table 3 the hybrid resin system will lead to a significant decrease in the tensile strength and tensile modulus of the composites. Please explain what is the advantages of hybrid resin system?

9# In the part of 2.3, the water absorption performance of composites is generally expressed by the percentage of water absorption content, so figure 5 should be changed.

10# Figure 6 is unprofessional. It should post process the original data to get a new figure.

11# Parts 2.5 and 2.6 should be compressive strength and flexural strength, not stress. In addition, please explain the evolution mechanism of compressive strength and flexural strength of three composites.

12# The conclusion must be further condensed, including only 3~4 key points, according to the current new findings.

Author Response

Dear Review 1

Based on your comments, we have made the following changes to the structure of the paper with

Title: “SOME MECHANICAL PROPERTIES OF COMPOSITE MATERIALS

WITH CHOPPED WHEAT STRAW REINFORCER AND HYBRID MATRIX”

Authors: Bolcu Dumitru, Stănescu Marius Marinel*, Mirițoiu Cosmin Mihai

Responses to specific comments. 

1# The current abstract does not convey some key information. Some qualitative and quantitative results on the mechanical properties, vibration properties, water absorption properties of composites are suggested to be presented in the current content. According to the findings of this research work, it can further promote the engineering applications of green composite.

Answer 1# We completed the abstract with some qualitative and quantitative information to justify the research carried out.

2# Introduction: this work aims to promote the engineering application of new green composite from agricultural waste in the field of industry as building materials. This idea is innovative, it can realize some advantages, such as waste utilization, ecological and environment protection, energy conservation and so on. Meanwhile, the above application of new green composite is also an attempt and challenge for synthetic fiber reinforced polymer composites, such as CFRP, GFRP, BFRP etc. As known, synthetic fiber composites have been widely used in the industrial field because of their excellent properties, mature preparation and application technologies. Therefore, to promote the new applications of green composites in the industrial field, this work should summarize some properties, advantages and application of synthetic fiber composites in the first part of introduction, and further put forward the main problems in terms of properties, cost, ecological environment to verify the future development potential of green composites in engineering applications. The authors can refer to the following latest work on performance, advantages and applications of synthetic fiber composites, such as Composite Structures, 2020; 246: 112418.  Construction and Building Materials, 2018, 161: 634-648.  Materials and Structures, 2020, 53: 73.

Answer 2# We reorganized the introduction and supplemented it with the bibliographic references indicated.

3# This paper studies the mechanical properties, microscopic properties, vibration behavior and water absorption properties of green composites. To be closely related to the current research content, the reviewer suggested that the authors should add the research on the vibration behavior and water absorption of green composites.

Answer 3# Some bibliographic references have been added in which the vibration behavior and water absorption of composite materials with hybrid resin matrices are studied.

4# There is a serious logical confusion in current writing in part 2. The experimental test should be separated from the results and discussion to form two independent parts. Part 2 should focus on the raw materials, sample preparation and mechanical property tests. Part 3 should be the results and discussion.

Answer 4# We have reorganized and completed the content of the article as requested.

5# Please explain why two hybrid resins systems are used in the experiment, and what advantages are there through the two hybrid resins? Will the two resin systems blend well? If it cannot be well blended, it is likely to form some initial defects owing to the poor interface.

Answer 5# We explained why the two hybrid resin systems were used and pointed out the main advantages of using these types of resins as matrices for casting composite materials.

6# Please provide some basic information about raw materials and sample preparation.

Answer 6# We have completed the information regarding the casting of composite materials and the methods of obtaining the samples.

7# Figure 1 is unclear; it is suggested to enlarge the key spectrum.

Answer 7# In figure 1, we marked with an outline the area where the chemical components are found. We enlarged this area and additionally added the respective area in the same figure (for a better visualization).

8# It can be found in table 3 the hybrid resin system will lead to a significant decrease in the tensile strength and tensile modulus of the composites. Please explain what is the advantages of hybrid resin system?

Answer 8# At the end of the results and discussion sections we highlighted the advantages of using hybrid resins and composite materials reinforced with chopped straw, which use these resins as a matrix.

9# In the part of 2.3, the water absorption performance of composites is generally expressed by the percentage of water absorption content, so figure 5 should be changed.

Answer 9# We expressed the variation of water absorption in percentages and replaced figure 5 (which after reorganization became figure 7).

10# Figure 6 is unprofessional. It should post process the original data to get a new figure.

Answer 10# We processed the original data and obtained a new figure showing how the frequency and damping factor are determined (figure 8).

11# Parts 2.5 and 2.6 should be compressive strength and flexural strength, not stress. In addition, please explain the evolution mechanism of compressive strength and flexural strength of three composites.

Answer 11# We have made the requested changes.

12# The conclusion must be further condensed, including only 3~4 key points, according to the current new findings.

Answer 12# We have condensed the conclusions.

We mention that the answers addressed to reviewer 1 are coloured in the text of the paper in blue, and the answers addressed to all reviewers are coloured in green (the corrected paper is in attachement).

Thanks for the views expressed on the basis of which we have made the changes that have contributed to increasing the scientific level of the paper.

Authors

Reviewer 2 Report

The paper seeks to introduce an approach ‘’ SOME MECHANICAL PROPERTIES OF COMPOSITE MATERIALS WITH CHOPPED WHEAT STRAW REINFORCER AND HYBRID
MATRIX”.
However, the authors should consider improving upon the quality to further highlight and emphasize. 

1.    State clearly in the abstract the results of the mechanical properties that were studied. An abstract should summarize the problem, what was done and achieved, and finally the significance of the study.

2.    In line 122, how was the specimen metalized with gold? What method was utilized? And how did you prevent any human error in doing so?

3.    The introduction needs to be improved by relating to the mechanics of the studied materials and their mechanical characteristics. The references to be included are: 10.1016/j.jiec.2022.06.023, 10.1177/0021998318790093, 10.1016/j.polymertesting.2017.09.009, 10.1016/j.compstruct.2021.114698 and 10.3390/polym14132662.

4.    It is very obvious that figure 1 was poorly represented. Kindly reinsert the figure with increased font size.

5.    Put space between each variable and its respective unit. In line 133, instead of 80 %, it is represented as 80%. Consider correcting throughout the manuscript if such exist.

6.    In lines 146 to 147, what did you use and how for cutting the specimen cutting?

7.    Align the x-y axes labels of Figures 3, 10, and 12 in the middle of their respective axis.

8.    Are the samples use an absolute number of the set of samples or an average?

9.    The magnification footers of figure 4 are not visible or blurred. Consider manually indicating inside the images.

10.What was the accelerating voltage, working range, and scale bar used during SEM analysis?

Author Response

Dear Review 2

Based on your comments, we have made the following changes to the structure of the paper with

Title: “SOME MECHANICAL PROPERTIES OF COMPOSITE MATERIALS

WITH CHOPPED WHEAT STRAW REINFORCER AND HYBRID MATRIX”

Authors: Bolcu Dumitru, Stănescu Marius Marinel*, Mirițoiu Cosmin Mihai

Responses to specific comments. 

  1. State clearly in the abstract the results of the mechanical properties that were studied. An abstract should summarize the problem, what was done and achieved, and finally the significance of the study.

Answer 1 We completed the abstract with some qualitative and quantitative information to justify the research carried out.

  1. In line 122, how was the specimen metalized with gold? What method was utilized? And how did you prevent any human error in doing so?

Answer 2 We completed the EDS analysis with the requested explanations.

  1. The introduction needs to be improved by relating to the mechanics of the studied materials and their mechanical characteristics. The references to be included are: 10.1016/j.jiec.2022.06.023, 10.1177/0021998318790093, 10.1016/j.polymertesting.2017.09.009, 10.1016/j.compstruct.2021.114698 and 10.3390/polym14132662.

Answer 3 We have reorganized the introduction and supplemented it with the bibliographic references indicated.

  1. It is very obvious that figure 1 was poorly represented. Kindly reinsert the figure with increased font size.

Answer 4 In figure 1 we marked with an outline the area where the chemical components are found. We enlarged this area and additionally added the respective area in the same figure (for a better visualization).

  1. Put space between each variable and its respective unit. In line 133, instead of 80 %, it is represented as 80%. Consider correcting throughout the manuscript if such exist.

Answer 5 We made the requested correction.

  1. In lines 146 to 147, what did you use and how for cutting the specimen cutting?

Answer 6 To cut the samples we used a water jet cutting machine (We added this information to the article).

  1. Align the x-y axes labels of Figures 3, 10, and 12 in the middle of their respective axis.

Answer 7 We aligned the x-y axis labels in Figures 3, 10 and 12 to the middle of each axis.

  1. Are the samples use an absolute number of the set of samples or an average?

Answer 8 The mean value and root mean square value was calculated for the number of samples of each set.

  1. The magnification footers of figure 4 are not visible or blurred. Consider manually indicating inside the images.

Answer 9 We corrected the error that appeared in figure 4 (became figure 6 after reorganization).

  1. What was the accelerating voltage, working range, and scale bar used during SEM analysis?

Answer 10 We completed the SEM analysis with the requested explanations.

We mention that the answers addressed to reviewer 2 are coloured in the text of the paper in red, and the answers addressed to all reviewers are coloured in green (the corrected paper is in attachement).

Thanks for the views expressed on the basis of which we have made the changes that have contributed to increasing the scientific level of the paper.

Authors

Round 2

Reviewer 1 Report

It is recommended to accept the current paper.